# Cytoskeleton, Transglutaminase and Gametophytic Self-Incompatibility in the Malinae (Rosaceae)

**DOI:** 10.3390/ijms20010209

**Published:** 2019-01-08

**Authors:** Stefano Del Duca, Iris Aloisi, Luigi Parrotta, Giampiero Cai

**Affiliations:** 1Dipartimento di Scienze Biologiche, Geologiche e Ambientali, Università di Bologna, Via Irnerio 42, 40126 Bologna, Italy; stefano.delduca@unibo.it (S.D.D.); iris.aloisi2@unibo.it (I.A.); luigi.parrotta@unibo.it (L.P.); 2Dipartimento di Scienze della Vita, Università di Siena, Via Mattioli 4, 53100 Siena, Italy

**Keywords:** self-incompatibility, pear, cytoskeleton, transglutaminase, pollen tube

## Abstract

Self-incompatibility (SI) is a complex process, one out of several mechanisms that prevent plants from self-fertilizing to maintain and increase the genetic variability. This process leads to the rejection of the male gametophyte and requires the co-participation of numerous molecules. Plants have evolved two distinct SI systems, the sporophytic (SSI) and the gametophytic (GSI) systems. The two SI systems are markedly characterized by different genes and proteins and each single system can also be divided into distinct subgroups; whatever the mechanism, the purpose is the same, i.e., to prevent self-fertilization. In Malinae, a subtribe in the Rosaceae family, i.e., *Pyrus communis* and *Malus domestica*, the GSI requires the production of female determinants, known as S-RNases, which penetrate the pollen tube to interact with the male determinants. Beyond this, the penetration of S-RNase into the pollen tube triggers a series of responses involving membrane proteins, such as phospholipases, intracellular variations of cytoplasmic Ca^2+^, production of reactive oxygen species (ROS) and altered enzymatic activities, such as that of transglutaminase (TGase). TGases are widespread enzymes that catalyze the post-translational conjugation of polyamines (PAs) to different protein targets and/or the cross-linking of substrate proteins leading to the formation of cross-linked products with high molecular mass. When actin and tubulin are the substrates, this destabilizes the cytoskeleton and inhibits the pollen-tube’s growth process. In this review, we will summarize the current knowledge of the relationship between S-RNase penetration, TGase activity and cytoskeleton function during GSI in the Malinae.

## 1. Introduction: The Molecular Basis of S-RNase-Based Gametophytic Self-Incompatibility in the Malinae

In plants, from the less to the more complex species, the reproductive process can take various forms, from the vegetative (asexual) one, which ensures the spread of individuals without genetic changes, up to sexual reproduction. In the latter case, the crossing of genetically different individuals produces a progeny that will be genetically different on both the maternal and paternal sides. The evolution of seeds, pollen, and then flowers in angiosperms has greatly facilitated this task with the help of both biotic (such as insects) and abiotic (such as wind) pollination vectors. The continuous interconnection between plants, animals and physical agents is probably one of the basic features of the extraordinary success of flowering plants [1]. This success has been facilitated by bisexual flowers, which carry both the male and female reproductive organs. While this has facilitated the reproductive process, it has raised the problem of self-fertilization, which might minimize genetic variability. In order to prevent or decrease the probability of this scenario, plants have adopted different strategies, from the temporally asynchronous development of male or female reproductive organs, to their specific positioning within the flower, up to strategies based on genetics [2,3]. In the latter case, we can use the term “self-incompatibility” (SI), a process that prevents inbreeding, thus promoting outcrossing. Mechanisms that prevent self-fertilization, such as the SI, also affect quantitative and qualitative aspects of productivity in agricultural species, many of which are of great economic interest. 

SI involves a pollen–pistil interaction and a cell–cell recognition system, which regulates the acceptance or rejection of pollen landing on the stigma of the same species, so that SI pollen arrests its tube elongation at a specific stage during pollination, preventing self-fertilization. A single locus controls most of SI systems, the so-called “*S* locus” which presents multiple *S*-alleles, i.e., the pistil-*S* and pollen-*S* genes; however, the process of pollen acceptance/rejection involves various other genes, many of which still need to be identified, and many other external factors [4,5]. Up to date two major classes of SI are known at the genetic level: the gametophytic SI (GSI) and the sporophytic SI (SSI). GSI is so named because the incompatibility phenotype of the pollen is determined by its haploid (gametophytic) genome exposed only after pollen grain’s germination, whereas with SSI the pollen exhibits the incompatibility phenotype of its diploid (sporophytic) parent [6]. Among the GSI systems studied to date, two are better characterized, one has so far been found only in the Papaveraceae [7], whereas the other GSI is present in various angiosperm families and has been extensively studied in the Solanaceae, Plantaginaceae, and Rosaceae [8].

As a comprehensive and exhaustive discussion of the mechanisms leading to SI would require much more space, here we focus on the GSI system, which can be subdivided into distinct mechanisms according to the species analyzed. More specifically, we will review the GSI system of Malinae, as in this Rosaceae subtribe both the molecular and the cytoskeletal events relating to the cytoskeleton and to TGase have been studied in the last decades. The timing of the events and the existing relationships mirror the complexity of the GSI process. In details, in this manuscript we focus mainly on the role of the cytoskeleton and transglutaminase (TGase) during the GSI response in Malinae. We will first briefly describe SI and the process of pollen tube development and how SI inhibits its growth, by altering TGase activity and cytoskeleton organization.

In Rosaceae, GSI generally blocks the pollen-tube growth at the level of the upper third of the style. It is widely accepted that the stylar *S* locus, encoding for glycoproteins showing ribonuclease (S-RNase) activity is the main player in this process. These RNases represent the female determinant that penetrate the pollen tube and are inactivated and degraded in compatible pollen, allowing the pollen tubes to grow [9]. On the contrary, during SI, the pistil’s S-RNases inside the pollen tubes are not degraded and the degradation of pollen’s RNA takes part in determining the arrest of the pollen tube’s growth. The pollen-*S* genes encoding for the male determinant are F-BOX proteins, named *S* locus F-Box Brothers (SFBB) in the subtribe Malinae of the family Rosaceae and S haplotype-specific F-box (SFB) in *Prunus* species of the family Rosaceae. Each SFBB protein seems to recognize a specific isoform of “non-self” S-RNase: in each *S*-haplotype different classes of SFBB proteins are thus essential for the degradation of “non-self” S-RNase and for the growth of the pollen tube [9,10].

While it is widely accepted that S-RNases degrade pollen RNA during SI, thereby preventing fertilization, it is likely that degradation of the target RNA is not the only mechanism by which self-pollen is rejected, and recently, an alternative mechanism differing from RNA degradation was proposed to explain the cytotoxicity of the S-RNase apple SI process [11]. In the apple SI response, S-RNase was shown to noncompetitively inhibit the specific pyrophosphatase activity of a soluble inorganic pyrophosphatase (MdPPa), leading to inorganic pyrophosphate accumulation, inhibition of tRNA aminoacylation and finally to PCD [11]. RNA degradation and MdPPa inhibition are not the solely mechanisms by which pollen tube growth is inhibited during SI. Other processes, including changes in cytoplasmic Ca^2+^ concentration and actin filaments (AFs) alteration, are likely to be involved [12]. This Ca^2+^-signaling cascade leads downstream to pollen tube tip growth arrest, depolymerization of AFs and consequently programmed-cell death (PCD) [13]. Both pollen-style interaction and SI response occur with different mechanisms in different plant families; however, since cytoplasmic Ca^2+^ concentration and the proper organization of the cytoskeleton are crucial for pollen tubes elongation, they might be common targets in mediating the SI process. On the other hand, as both Ca^2+^ fluxes and cytoskeleton dynamics are involved in vesicle trafficking, they might regulate the uptake of S-RNase. A fine interconnection among cytoplasmic Ca^2+^ concentration, AFs organization and ROS is of great importance for the process of SI in pear pollen, and slight alterations of this balance, induced for example by the supplementation of apoplastic calmodulin, can block the SI process [14]. Evidence supporting the role of the cytoskeleton during SI response in Malinae may also come from the analysis of SI processes occurring in other systems. For example, in poppy (*Papaver rhoeas*), which has a GSI system not based on S-RNase. In this SI system, the female determinant is a protein (PrsS) secreted to the pistil surface and the male determinant is a transmembrane protein (PrpS) located in the pollen. In this system, SI pollen recognition causes an increase of intracellular Ca^2+^, which induces a multi-layered SI signaling cascade that culminates in PCD of incompatible pollen grains [15].

The dynamics of AFs are crucial in the regulation of PCD and changes in AFs trigger a caspase-like protease pathway. The application of drugs inhibiting the cytoskeleton organization also highlight the importance of AFs as these accelerate the onset of the SI response [16]. Microtubules (MTs) are a chronologically later target of SI response, since SI triggers rapid depolymerization of MTs only after actin depolymerization; alleviation of SI-induced PCD by taxol-stabilized MTs suggests that MTs depolymerization also mediates PCD [17]. The above examples come from a different SI system and should therefore be taken with caution when trying to adapt them to the Malinae system; however, they suggest that the cytoskeleton of pollen tubes (both AFs and MTs) may be part of the complex mechanism that regulates the PCD progress during SI response. 

## 2. Pollen Tube Growth and How It Is Impacted by Self-Incompatibility

The cytoskeleton plays an important role in the development of pollen tubes as it promotes organelle and vesicle motility and facilitates the correct structuring of the cell wall. AFs are generally organized into three arrays: short filaments at the apex, the actin fringe in the sub-apex, and longitudinal actin strands in the rest of the cytoplasm. The dynamics and function of MTs in the pollen tube is not yet well defined [18,19]. As in other plant cell types, it is reasonable to think that the two cytoskeletal systems interact by affecting their reciprocal dynamics and by cooperating in the localization and distribution of organelles and secretory vesicles. In the latter case, AFs determine their fast and long-distance transport while MTs are likely involved in their local and precise positioning [20]. Secretory vesicles contain all the material necessary for pollen tube growth, which occurs mainly at the apex [21] and mainly depends on vesicular secretion that occurs very precisely at the tip. This is essential to convey new plasma membrane, methyl-esterified pectins and proteins that are used for the building of the new cell wall [22,23,24]. The subsequent conversion of methyl-esterified pectins into acidic pectins and their Ca^2+^-dependent cross-linking stiffens the cell wall thereby slowing down cell growth. The alternation between cell wall relaxation and stiffening, together with the action of turgor pressure, determines the so-called “pulsed” or “oscillatory” growth of pollen tubes. This sequence of events is repeated regularly and implies oscillation of various molecular mechanisms, including Ca^2+^, ROS and GTPases [25]. The cytoplasmic influx of high amounts of Ca^2+^ might alter ROS concentration because NADPH oxidase is directly regulated by Ca^2+^ [26]. ROS are extremely reactive chemicals containing oxygen, as peroxides, superoxide, hydroxyl radicals, and singlet oxygen. ROS are formed as a natural product of oxygen metabolism and play an important role in cellular signaling and homeostasis. During stress conditions, ROS levels can increase significantly, causing damage to cell structures (oxidative stress) [27]. In the specific case of SI response in *Pyrus pyrifolia*, suppression of the growth of SI pollen tubes in pear also involves changes in ROS levels, probably linked to variations in intracellular Ca^2+^ [14,28]. An interesting fact also arises from the evidence that ROS levels increase in pear pollen tubes grown in the presence of polyamines [29]. Excess of ROS can affect the activity of proteins in the apical region and modifies the structure of AFs [30,31]. The latter event is consequently responsible for changes in the transport and in the accumulation of secretory vesicles into the apex. 

It is widely accepted that the growth of pollen tubes is inhibited by S-RNase degrading the RNA of incompatible pollen tubes. However, Li and coworkers demonstrated that the arrest of apple pollen growth was dependent also on the inhibition of pollen soluble pyrophosphatase activity, indicating that RNA degradation may not be the only mechanism required for pollen rejection in apple. The identified soluble inorganic pyrophosphatase (MdPPa) was highly expressed in germinated pollen and interacted with S-RNases, both in vivo and in vitro assays, suggesting that it plays a role in the apple SI response. MdPPa was identical in several apple cultivars and its activity decreased in SI-treated pollen tubes, suggesting that MdPPa is involved in the SI response [11].

Moreover, in *Pyrus*, S-RNase had several different effects on pollen tubes during SI induction in vitro and altered the fine modulation of ROS and Ca^2+^. S-RNase disrupted tip-localized ROS and decrease the Ca^2+^ current and, finally, degraded the nuclear DNA. The authors speculated that S-RNase simultaneously decrease Ca^2+^ current and inhibit ROS formation; however, as Ca^2+^ currents and ROS production are mutually interconnected, it is difficult to distinguish the order in time between tip-localized ROS disruption and Ca^2+^ current decrease [28]. Cytoplasmic Ca^2+^ changes drastically affected the dynamics of AFs and MTs [13,28]. If AFs are not arranged correctly, this causes improper secretion and thus results in deeply altered cell wall deposition. If the amount of ROS increases in an uncontrolled manner [32], it can cause both toxic effects and an imbalance in other components such as cytoplasmic Ca^2+^. Taken together, S-RNase in Malinae may participate in a more complicated pathway, such as PCD, following its secretion from the style to its ultimate function in inhibiting self-pollen growth.

## 3. The TGase-Cytoskeleton Interplay as a Crucial Player of the SI Process

TGases are a class of enzymes that catalyze the post-translational modification of proteins by the formation of isopeptide bonds either through protein cross-linking via ε-(γ-glutamyl) lysine bonds or through incorporation of primary amines. The resulting cross-linked products often show high molecular mass [33]. TGases are widespread, from fungi to algae and higher plants, and are distributed in different cell compartments (cell wall of algae and fungi, chloroplasts, and mitochondria) where they exert structural or conformational roles [34]. In plants, much effort has been put in studying the post-translational modification of proteins by polyamines (PAs) [35,36]. Plant TGases show little sequence homology with the animal counterparts, except for the catalytic amino acid triad [37,38].

In pollen, TGase exists in at least two distinct forms, one is cytosolic, and one is associated with the cell wall. It was demonstrated in pear pollen tube that TGase is present in different subcellular compartments and that its distribution is regulated by both membrane dynamics and cytoskeleton integrity, suggesting that delivery of TGase to the cell wall requires the transport of vesicles containing membrane components along cytoskeleton filaments. Information on the distribution of TGase in pear pollen tubes was obtained using immunofluorescence and immunogold electron microscopy [39]. Visualization of whole pollen tubes showed that TGase was distributed in the growing half, suggesting that the enzyme is actively constrained to accumulate in the growing segment, possibly achieved by the association of TGase with intracellular membranes. TGase is also to be associated to the plasma membrane of pollen tubes, indicating that the enzyme might be either secreted or deposited to locally modulate the cell wall structure [39] and probably to take part in the modulation of signal transduction. The distribution of TGase in pear pollen tubes was shown to be comparable to that found in apple pollen tubes [40], suggesting that TGase might play an important role at the interface between the pollen tube and stigma/style cells in Malinae. In order to determine the contribution of membrane dynamics to the distribution of extracellular TGase, the effects of inhibitors of cytoskeleton organization and membrane transport have been investigated. Results suggest that inhibitors of cytoskeleton (Latrunculin B for actin, and oryzalin/taxol for tubulin) appreciably affected the localization of TGase in pear pollen tubes, preventing the accumulation of TGase in the growing apical domain. Simultaneously, application of either Latrunculin B or Brefeldin A, an inhibitor of vesicle transport, resulted in the accumulation of TGase in the endomembrane compartments of pear pollen tubes with a lower abundance of TGase in the cell wall [39]. Brefeldin A is a fungal toxin that inhibits the transport of vesicles [41] by inhibiting a GEF (guanine nucleotide exchange factor) associated with Golgi and blocking the activation of ARF1 (ADP-ribosylation factor 1) and the assembly of COPI (coat protein complex) on Golgi membranes.

Collectively, these data indicate that delivery of TGase to the cell wall of pear pollen tubes is dependent on both cytoskeleton and membrane dynamics. Data showed that cytoplasmic TGase interacts with the cytoskeleton, while a different TGase isoform, probably delivered via a membrane/cytoskeleton-based transport system, is secreted in the cell wall of pear pollen tubes, where it might play a role in the regulation of apical growth [39] during SI [42]. The two forms are probably involved in different functions, with the cell wall-associated TGase being involved in the apical growth of pollen tubes. This assumption is based on evidence that both TGase-specific inhibitors and an anti-TGase monoclonal antibody can block apple pollen tube growth, while the incorporation of a recombinant fluorescent TGase substrate into the pollen tube wall enhances pollen tube germination and elongation [43]. These findings suggest a possible role for TGase in mediating the interactions between the pollen tube and the style during pollen tube elongation in apple [40]. 

Although TGases are of great importance for pollen tubes growth, enhanced enzyme activity appears to be required also during the SI response [42,43] and PCD-related processes, such as those occurring in the flower corolla [44]. The stimulation of TGase activity during the PCD pathway is mainly due to the increase of Ca^2+^ (i.e., by releasing Ca^2+^ from cellular storage compartments and/or by external influx of Ca^2+^). This is a typical TGase feature in different cellular systems studied so far [33,34,38,45]. Not only TGase activity significantly changes during SI in pear pollen, but also the content of PAs, well-known TGase substrates [42], leading to the hypothesis, that TGase could act in the pathway leading to cell death during the SI response. Such a change in the role of TGase may be related to modifications of the cellular environment in which the enzyme operates, such as changes in Ca^2+^ levels or of intracellular PAs.

The above evidence collectively suggests that TGase is a regulator of pollen tube growth and that it might be a critical mediator of the SI mechanism, although molecular details of this process are still missing. Recently, cytoplasmic TGase of apple pollen was demonstrated to catalyze the PA-based post-translational modification of actin and tubulin, generating aggregates of high molecular weight. TGase-induced modification of the cytoskeleton proteins also affected the binding affinity of the motor proteins myosin and kinesin [46] probably influencing cytoplasmic dynamics based on cytoskeleton motor proteins (i.e., organelle transport). Both the cytoplasmic and the cell wall-associated TGases are involved in this complex scenario, being key regulators of cytoskeleton and cell wall dynamics respectively. In pear, TGase activity is stimulated in compatible styles and high molecular mass cross-linked products are formed during the SI response [47]. In vitro experiments conducted with purified TGase, actin and tubulin, showed that inhibition of pollen tube growth in SI crosses is mediated by an abnormal reorganization of the cytoskeleton [48,49]. In *Citrus* (a putative S-RNase-based SI) abnormal tube morphology was observed during SI, with callose deposition in the tube wall and apex. An increase of PCA-soluble and PCA-insoluble PAs (the latter including TGase-conjugated PAs) also occurred with a peak in concomitance with the arrest of pollen tube growth. TGase activity in *Citrus* also increased during SI pollination, while compatible pollination showed a decrease in enzyme activity. In addition, glutamyl-PA conjugates reached a maximum in SI-pollinated pistils concomitantly with the cessation of pollen tube growth [43].

In pear pollen, it was observed that both TGase activity and PA content change during SI [42], indicating the importance of TGase during SI response. Moreover, also the distribution of pear TGase was altered during SI, and the enzyme appeared distributed in patches along the pollen tube [39]. Therefore, the SI response affected also the localization of TGase in growing pollen tubes.

The increase of intracellular Ca^2+^ concentration as observed during SI response in pear pollen tubes [32] might cause the stimulation of TGase activity in SI pollen tubes. Once activated, TGase might disorganize the pattern of AFs and MTs by post-translational linkage of PAs or by crosslinking protein substrates, as it does in vitro [46], leading to the disrupting effects observed in pear pollen [12]. Abnormal reorganization of AFs induced by TGase might be similar to the so-called actin foci, atypical actin structures observed during SI response in poppy [50], and to which specific AF binding proteins take part. Although actin/tubulin aggregates and actin foci are likely different structures as they are part of different SI systems, both could affect the growth rate of pollen tubes. In the case of the S-RNase-based SI system of Malinae, one consequence of actin disorganization would be changes of vesicle trafficking, which would impair the deposition/secretion of extracellular TGase. 

Since precise levels of cell wall-bounded TGase are required for the correct growth of pollen tubes [40], the increase of TGase activity during the SI might impair the growth process leading to PCD [5,12,28]. The hypothetical pathway linking SI to TGase activity is reported in Figure 1. In summary, current data suggest that the SI response likely involves either the up-regulation or down-regulation of TGase activity, which may in turn affect the organization and function of the cytoskeleton and consequently the deposition of cell wall-associated TGase, indicating this enzyme as a key player during the SI response.

The next question is how TGase interfaces with cytoskeletal activity. A couple of hypotheses can be made, one involving a direct relationship, while the other involves an indirect relationship. In the first hypothesis, TGase affects the structure and thus the activity of the cytoskeleton. TGase is a Ca^2+^-dependent enzyme and the SI response can alter Ca^2+^ levels in SI pollen tubes. The onset of a SI response could significantly alter Ca^2+^ levels and thus compromise TGase activity. The latter could modify actin and tubulin by post-translational modifications and contribute to the formation of unusual structures. We cannot correlate the in vitro assembly of actin and tubulin aggregates to the actin foci observed in other SI systems; their nature is likely different, but they could intriguingly have the same purpose, that is altering the cytoskeleton structure to promote SI. In the second hypothesis, TGase may have an indirect effect on the cytoskeleton by altering the structure of the cell wall. Consequently, an altered growth rate would affect the cytoskeletal organization and be responsible for abnormal cytoskeletal structures. We cannot affirm that the actin foci observed in other types of SI are caused by the altered activity of TGase; what we can instead affirm is that aggregates of actin and tubulin induced in vitro by TGase [46,49] could also be replicated in vivo. Direct or indirect, the relationship between TGase and the cytoskeleton is certainly part of the S-RNase-based growth control mechanism of pollen tubes.

In both apple and pear, the SI response certainly involves the cytoskeletal apparatus in a way that is not yet fully clarified. Yang and coworkers recently studied the timing of the events affecting the cytoskeleton during the SI response in apple; they demonstrated that AF depolymerization takes place within 6 minutes after SI induction, while pollen tube MTs are disrupted and form foci only 60 min after SI induction [13,51]. Therefore, MTs are likely to be a secondary target of the SI process but could also play more roles that are important during the SI response. In S-RNase-based SI systems, such as apple, stabilization of MTs with taxol delays cell death, suggesting that depolymerization of MTs is a necessary step to induce pollen tube death. Nevertheless, it is also true that stabilization of MTs can slow down the penetration of S-RNase [52]; therefore, MTs are both a mediator of the SI response but also one of the targets. In the light of these partial data, it could be speculated that depolymerization of MTs, following the SI response, is merely a consequence of cell death and, therefore, a non-specific event. It is also conceivable that MTs may play an active role in the penetration of S-RNase and therefore their role may be more important during the initial phase of the SI response. On the other hand, the involvement of MTs in endocytosis has already been suggested [53] and this would strengthen the hypothesis that they are needed to capture and internalize S-RNase.

The link between S-RNase internalization and disruption of the cytoskeleton may involve at least two other components, i.e., cytoplasmic Ca^2+^ and ROS [14,28,54]. As mentioned above, the concentration levels of cytoplasmic Ca^2+^ fluctuate during pollen tube growth so that it is synchronous to other events. It is believed that the increase in cytoplasmic Ca^2+^ concentration follows the fast-growing phase. Data on changes in cytoplasmic Ca^2+^ during S-RNase-based SI are often conflicting. In the pear system, intake of S-RNase may cause a decrease in intracellular Ca^2+^ levels probably by acting on the permeability of the plasma membrane or calcium channels [55]. It is believed that intake of self S-RNase may inhibit the activity of phospholipase C associated with the plasma membrane, which in turn converts phosphatidylinositol 4,5-bisphosphate (PIP_2_) into inositol 1,4,5-trisphosphate (IP_3_), an effector capable of opening calcium channels [4]. However, intake of S-RNase may also decrease the apical concentration of ROS [32]. Since the concentrations of cytoplasmic Ca^2+^ and ROS are interrelated [56], the effect of S-RNase could only be exerted on cytoplasmic Ca^2+^ and then be reflected on altered ROS levels. In *Nicotiana alata*, (another S-RNase-based SI system) it was observed that disorganization of AFs precedes the damage to the vacuolar system; the latter is the container of the S-RNase that penetrated the pollen tubes. Consequently, the damage to AFs occurs before the S-RNase is released into the cytoplasm [57,58]. It is not clear whether S-RNase directly affects the structure of AFs, when this may happen and if actin is an indirect or direct target of S-RNase. Although in vitro biochemical tests have shown that S-RNase does not bind or damage filamentous actin, in apple, S-RNase interacts with an actin-binding protein (MdMVG) that binds and severs F-actin. Specifically, S-RNase inhibits the F-actin-severing activity of MdMVG. thereby modifying the dynamics of AFs and reducing pollen tube growth [13]. The above data suggested that the alteration of AFs can occur at least in two ways: on the one hand S-RNase could act directly on proteins that regulate actin dynamics, on the other hand the effect of S-RNase could be achieved by altering factors such as Ca^2+^, which in turn affect the activity of actin-binding proteins or activate various enzymes, such as TGase, for which actin is one target.

## 4. Future Perspectives

Although the process of SI has been studied for decades, many aspects remain to be defined, including the one concerning the involvement of the cytoskeleton. The fundamental question is whether the cytoskeleton is a target of pollen tube growth arrest or whether it is an active part of the rejection process. A second important aspect to be defined is the link between the actual onset of incompatibility (hence the entry of RNase) and the modifications induced to the cytoskeleton. In this context, the role of TGase seems to be crucial, as it modifies post-translationally both actin and tubulin. Defining the role of TGase might require several biochemical and molecular approaches. Certainly, the overexpression of TGase in compatible and SI pollen would be of great interest to verify how crucial cytoskeleton modifications are during the SI response. 

## Figures and Tables

**Figure 1 ijms-20-00209-f001:**
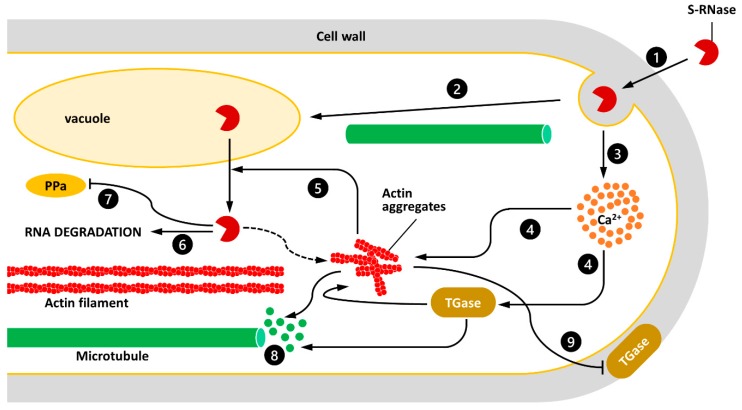
A speculative model illustrating the interactions between S-RNase, TGase, and the cytoskeleton of pollen tubes. (1) S-RNase is internalized through endocytotic mechanisms. (2) MTs are likely to be involved in the delivery of S-RNase to the vacuole. (3) Intake of the S-RNase results in an increase in the cytosolic concentration of calcium ions. (4) Increasing the Ca^2+^ concentration can affect the stability of AFs and unbalance the activity of TGase. (5) The alteration of AFs promotes the release of S-RNase from the vacuole. (6) S-RNase may degrade the RNA of pollen tubes; (7) alternatively, S-RNase may inactivate specific enzymes such pyrophosphatases (PPa). (8) The depolymerization of AFs or the formation of actin aggregates might in turn destabilize MTs and related processes. (9) Damage to the cytoskeleton affects the secretion of the extracellular TGase, which in turn affects pollen tube growth.

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
