# Peer review of "Cytoskeleton, Transglutaminase and Gametophytic Self-Incompatibility in the Malinae (Rosaceae)"

_ijms, 2019, doi:10.3390/ijms20010209_

Round 1

Reviewer 1 Report

The manuscript just needs a few minor corrections in  Line 146 (tubs??) and Line 272 (electing??)

Author Response

We thank the reviewer for the suggestions. We have incorporated the required changes in the new manuscript version.

Reviewer 2 Report

Please see the attached PDF file marked by "track-changes" that I have sent to the Editor separaetly.

Author Response

We thank the reviewer very much for his/her kindness in the correction of the text, it was appreciated so much! Suggestions of modification have been incorporated in the new version. In addition, we also rearrange some parts of Section 3 in order to make concepts more linear.

Reviewer 3 Report

The authors depart from the orthodoxy of gametophytic self-incompatibility (GSI), which tends to focus on S-RNase/F-box and a handful of pistil proteins (120kDa and HT-A/B) as the chief determinants of GSI in the Solanaceae, Plantaginaceae, and Rosaceae. They focus instead on calcium dynamics, the cytoskeletal framework, ROS, and some unexpected protein types (pyrophosphatases and TGases) as not only key players in the SI response in apple and pear but as causative agents in programmed cell death of incompatible pollen tubes. 

There are some inconsistencies in their argument. For example, in Line 149 they state that S-RNase is possibly not required for SI in the Malineae, but the model in Figure 1 not only rests on, but begins with,  the assumption that S-RNases are the causative agents in the arrest of incompatible pollen tubes.  Also, if we accept the latter notion--and there is certainly a much larger body of evidence in support of the role of S-RNases in GSI, including in the Malineae--it follows that progressive RNA degradation will produce a great deal of collateral damage, including disruption of calcium dynamics and the cytoskeletal network, and metabolic upheaval that could include temporary upregulation of TGases (because some inhibitor of these enzymes has been annihilated during the RNA armageddon). In fact, it stands to reason that stable proteins, which by definition can withstand loss of their RNA pool, will be at an advantage at least for a while, in such a scenario.  Another troubling aspect of some of the studies that the authors quote in support of their model is that they are based on pollen tubes grown in vitro, an assay system that is demonstrably ineffective in replicating the in situ GSI response; for example, S-RNases act in a non-S-allele-specific manner to inhibit pollen tube growth in vitro. Finally, if RNases destroy the RNA in an incompatible pollen, isn't any additional inhibitory effect overkill? Pardon the pun.

Despite these reservations, I support publication of this review because I believe it strengthens scientific progress when alternative views are aired and debated. My only suggestion is that the authors attempt to make their model internally consistent, explain why other inhibitory effects (invoking the cytoskeleton or TGases) are necessary if the RNA pool in the cell is lost, and also what role the S-locus F-box protein might play in their model. I do agree with the authors that there is a great deal about the self-incompatibility response that needs to be clarified at the molecular level.

Author Response

We thank the reviewer for the comments and suggestions. Indeed, the statement to which the reviewer refers suggests that S-RNase may not be necessary to trigger the SI response. This statement must be rectified; we have therefore softened the meaning of the sentence (highlighted in yellow). The data in the literature do not allow to assert that S-RNase is not required for SI in the Malineae, but that S-RNase can act in different ways. For example, S-RNase could also modulate pyrophosphatase activity. This hypothesis has now been inserted in the model of figure 1.

The second point raised by the reviewer concerns the effects of the progressive degradation of RNA.  It is logical to think that RNA degradation can cause numerous effects, including cytoskeletal damage and metabolic changes (including alterations to TGase activity). Nevertheless, the exact sequence of events is not known, and some events may occur even earlier than RNA degradation. The lack of a precise timeline is also dependent on the evidence that some data come from in vitro studies, while others from in planta studies. It is true that the in planta system is more complete but the in vitro system allows to modulate certain factors and to analyze others more precisely. Unfortunately, this does not allow the events to be aligned according to a precise sequence. The data in literature, not always in agreement with each other, do however suggest that other events, besides RNA degradation, are necessary or consequential. The major difficulty still existing today is the distinction between a "necessary" or "consequential" event. Some parts of Section 3 have been changed in order to “linearize” the concept; changes have not been highlighted because they are essentially rearrangement of text blocks.

As for the role of the S-locus F-box protein, we have some concerns about introducing it into the proposed model. The subject of the review is focused on the effects induced by self S-RNase and on the events that may occur in cascade. The discussion on F-Box and the relationship with other structures, such as cytoskeleton and TGase, would perhaps require a further specific article. We prefer not to discuss F-box in this review and we hope that the reviewer will understand our point of view.